# Moisture Resistance, Thermal Stability and Fire Behavior of Unsaturated Polyester Resin Modified with L-histidinium Dihydrogen Phosphate-Phosphoric Acid

**DOI:** 10.3390/molecules26040932

**Published:** 2021-02-10

**Authors:** Kamila Sałasińska, Maciej Celiński, Kamila Mizera, Mateusz Barczewski, Paweł Kozikowski, Michał K. Leszczyński, Agata Domańska

**Affiliations:** 1Central Institute for Labour Protection—National Research Institute, Department of Chemical, Biological and Aerosol Hazards, 00-701 Warsaw, Poland; macel@ciop.pl (M.C.); kamiz@ciop.pl (K.M.); pakoz@ciop.pl (P.K.); 2Faculty of Materials Science and Engineering, Warsaw University of Technology, 02-507 Warsaw, Poland; 3Institute of Materials Technology, Poznan University of Technology, 61-138 Poznań, Poland; mateusz.barczewski@put.poznan.pl; 4Faculty of Chemistry, Warsaw University of Technology, 02-507 Warsaw, Poland; mleszczynski@ichf.edu.pl; 5Institute of Physical Chemistry, Polish Academy of Sciences, 01-224 Warsaw, Poland; 6Łukasiewicz Research Network—Institute for Engineering of Polymer Materials and Dyes, 87-100 Toruń, Poland; agata.domanska@impib.lukasiewicz.gov.pl

**Keywords:** unsaturated polyester resin, burning behavior, fire retardant, thermal stability

## Abstract

In this paper, the fire behavior of unsaturated polyester resin (UP) modified with L-histidinium dihydrogen phosphate-phosphoric acid (LHP), being a novel intumescent fire retardant (IFR), was investigated. Thermal and thermomechanical properties of the UP with different amounts of LHP (from 10 to 30 wt. %) were determined by thermogravimetric analysis (TG) as well as dynamic mechanical thermal analysis (DMTA). Reaction to small flames was studied by horizontal burning (HB) test, while fire behavior and smoke emission were investigated with the cone calorimeter (CC) and smoke density chamber. Further, the analysis of volatile products was conducted (TGA/FT-IR). It was observed that the addition of LHP resulted in the formation of carbonaceous char inhibiting the thermal decomposition, burning rate and smoke emission. The most promising results were obtained for the UP containing 30 wt. % of LHP, for which the highest reduction in maximum values of heat release rate (200 kW/m^2^) and total smoke release (3535 m^2^/m^2^) compared to unmodified polymer (792 kW/m^2^ and 6895 m^2^/m^2^) were recorded. However, some important disadvantage with respect to water resistance was observed.

## 1. Introduction

Unsaturated polyester resins (UPs) are among the most vital thermoset resins used in versatile areas, especially transportation, construction, chemical equipment, electrical installation and decoration. This is mainly due to their attractive properties, including prominent mechanical performance, low cost, solvent resistance, and dielectric properties [1,2,3]. UPs are also widely employed as adhesives and coating on high-end furniture and musical instruments due to their high gloss and perfect fullness [4,5]. UP is usually a polyester solution (60–70%) containing unsaturated bonds in a cross-linking monomer. The resins’ curing occurs as a radical reaction initiated by thermal or photochemical factors; however, processes using organic peroxides or hydroperoxides currently prevail due to higher costs. Vinyl, allyl and acrylic monomers are commonly used in a copolymerization reaction with unsaturated polyester, but the most commonly used in the industry is styrene [3]. The presence of a low molecular weight monomer allows the increase of the degree of cross-linking of the resin and the decrease of the viscosity. Such behavior enables the use of UP as a matrix of composite materials. It has been proven that the degree of cross-linking, which determines mechanical properties, as well as chemical resistance of the material, depends on the chemical composition of the polyester (the number of double bonds) and the amount of cross-linking monomer [4]. The high content of flammable styrene causes a high risk of fire [6,7].

The problem of flame retardancy of polyester resins is currently one of the most important and most difficult tasks for modern polymer technologies [8]. To improve the combustion resistance of unsaturated polyester resin, flame retardant (FR) additives have been widely used. Commercial flame retardants, which are usually incorporated into the polymer during the processing, contain halogens such as bromine [9] or chlorine, phosphorus, nitrogen [10,11] or metal hydroxides like aluminum and magnesium [12]. Moreover, during the burning of unsaturated polyester resins, significant amounts of dense smoke are emitted. This phenomenon is accompanied by the emission of toxic gases (HCl, HBr, CO), especially for UP resins containing halogen fire retardants. For that reason, intumescent flame retardants (IFRs), inhibiting the combustion process by creating a swollen char, are gaining more and more popularity nowadays.

Considering the wide application and low fire resistance of UP, the problem of reducing their flammability has been raised in many publications. Kandre et al. [13] have developed a system containing ammonium polyphosphate (APP) and zinc-based smoke suppressants (zinc stannate (ZS), zinc hydroxystannate (ZHS) and zinc borate (ZB)). The combination of ZS, ZB, and ZHS with APP improved the polyester resin’s thermal stability at elevated temperatures and subsequently promoted the charring process. Chen et al. [14] used triphenyl phosphate (TPP) to microencapsulate diatomine/ammonium polyphosphate (Dia/APP) to improve the flame retardancy of UP. The incorporation of Dia-APP-TPP into UP greatly enhanced the flame retardancy and thermal stability of the polymer. UP with 20 wt. % of Dia-APP-TPP successfully achieved UL-94 V-0 rating, and the LOI value increased from 19.2% to 26.6%. Furthermore, the cone calorimeter test showed that the total heat release (THR) decreased by 33.9% compared to unmodified UP. Pan et al. [3] added ammonium polyphosphate (APP) and triphenyl phosphate (TPP) into UP and investigated the fire resistance effect. The addition of 57.6 wt. % of the flame retardant system caused an increase in LOI value from 20.9 to 27.2 and allowed to obtain a V-0 rating in the UL-94 test. In turn, Liu et al. [5] showed that only 15 wt. % of the mixture of dimelamine pyrophosphate (DMPY) with aluminum diethylphosphinate (ADP) incorporated into UP caused V-0 rating in vertical burning tests, and an increase in the limiting oxygen index to 27.9%. Moreover, the heat release rate (HRR) and effective heat of combustion (EHC) of UP/FRs were conspicuously suppressed, and UP/FRs retained satisfactory mechanical performance after water resistance tests. Novel and environmentally friendly flame retardant for UP was elaborated by Chen et al. [4]. They developed an effective intumescent flame retardant architecture based on chitosan (CH) and ammonium polyphosphate (APP), which were successfully deposited on the surface of diatomite particles by Layer-by-Layer (LbL) assembly. Cone calorimetry test reveals that IFR’s addition of reduces the peak heat release rate (pHRR) and THR of UP by 40.8% and 18.1%, respectively.

Most of the IFRs are based on phosphorus compounds. However, their use in polymeric materials is not without disadvantages; the more important is the lower thermal resistance and increased water absorption [15]. Thus, this study aimed to investigate the impact of developed, non-halogen flame retardant L-histidinium dihydrogen phosphate-phosphoric acid (LHP) on water resistance, thermal stability and flammability of unsaturated polyester resin. A few research methods, such as thermogravimetric analysis, horizontal burning test as well as cone calorimetry and smoke density chamber measurements, were used to define the impact of FR on the thermal stability and burning behavior of modified UP. The influence of the LHP was complemented by a microstructure analysis of the materials before and after cone calorimetry tests. Additionally, TGA combined with FT-IR was carried out to determine the substances, which could be evolved during the material’s thermal decomposition. The effect of the addition of flame retardant on the structure and thermomechanical properties was determined using dynamic-mechanical non-isothermal analysis carried out under the determined conditions of mechanical excitation. DMTA was performed for a UP containing 30 wt. % of the modifier subjected or not to soak in water. The results were compared to those obtained for UP and resin modified with commercial flame retardant.

## 2. Results and Discussion

### 2.1. Fire Retardant’s Characteristic

Histidine and LHP were tested using FT-IR to confirm the presence of all expected chemical groups and verify the presence of any side products. In the range of 800–1000 cm^−1^ multiple deformation vibrations peaks of ring C-N bonds were observed. In turn, stretching vibrations of C-C and C-N are present in the range of 1000–1300 cm^−1^. Moreover, in the range of 1400–1700 cm^−1^, peaks can be attributed to stretching vibrations of C=O and C-H as well as bending vibrations of NH_2_. For the LHP (red line in Figure 1), it was determined that the phosphoric acid shows stretching vibrations of P-O and P=O, which translates to absorbance in the wavelength range 800–1200 cm^−1^ and is clearly seen in the product spectrum. Some shifts can be observed for the amine groups in the bending vibrations of N-H (1400–1500 cm^−1^) and stretching vibrations of NH_2_ (2800–3000 cm^−1^), which is probably the effect of ionic bonding with the phosphoric acid.

Elemental analysis (CHN) was employed to evaluate the composition and purity of the synthesized LHP. As demonstrated in Table 1, the experimental data fits best to the theoretical composition of 96% of the pure LHP phase and 4% of histidine. This interpretation of the experimental data is corroborated by the PXRD of the product synthesized using the reported method, revealing the presence of a nearly phase-pure LHP [16].

### 2.2. Microstructure Analysis

The morphology of the UP modified with IFRs was studied by SEM, and the images are shown in Figure 2.

Analysis of SEM images of UP/APP allows observing the presence of differences in the size of FR’s particles and their random arrangement in the entire volume of resin. Moreover, the formation of agglomerates was noted. UP/LHP revealed a more uniform dispersion of FR particles, even in the case of samples with the highest share of LHP. The voids distributed in the polymer, resulting mostly from the particles’ loss or occasionally a hindered degassing of the highly viscous compositions, were observed. In UP/APP, the voids in the area of contact between the flame retardant particles and resin were also present (Figure 2c). There was no porosity resulting from the premature start of the swelling process, as an effect of elevated temperature during the materials’ manufacturing.

### 2.3. Thermal Stability and Water Resistance

The thermal stability and degradation behavior of UP with FRs were studied by TG analysis. The decomposition process details are summarized in Appendix A (supplementary data), while the TG and DTG curves are presented in Figure 3.

The unmodified unsaturated polyester resin exhibited the 5% of weight loss temperature (T_5%_) of 278 °C, which corresponds to the decomposition′s onset temperature. The addition of APP increased the T_5%_, and for most samples, the values grew as the share of FR increased. The opposite tendency was observed for resin modified with LHP, for which the increase in FR′s share resulted in a gradual reduction in the analyzed parameter. It was attributed to the accelerating decomposition of the organic compound by the presence of phosphoric acid in LHP. Shorter chains did not volatilize, and instead, they cyclized and formed thermally stable aromatic compounds co-creating the char. As shown in Appendix A, the residual mass of UP/20LHP and UP/30LHP was a few times higher than that of unmodified UP and the highest from all UP/IFRs samples.

The decomposition process of UP has two main steps. The first one at the temperature of 120–275 °C corresponds to water loss by dehydration. The main degradation step ranging from 275 to 450 °C results from the chain scission of polystyrene as well as polyester fragments [2]. The addition of intumescent fire retardants substantially altered the decomposition of UP. The first stage is ascribed to the dehydration, while the next two to the decomposition of FRs (no obvious peak for UP/20APP) and the polymer. The intensity of degradation shows that the decomposition rate of UP/IFRs was lower than that of the resin and decreased with the share of FRs increased. The slower decomposition was probably due to the char’s formation, which decreases the thermal conductivity of the materials and consequently increases its fire resistance. The degradation of transient char at a temperature above 550 °C, a characteristic mostly for the analysis carried out in oxygen conditions [2], can be observed. The decomposition of pyrolytic soot occurred as a small peak at 652 °C also for unmodified UP. In the case of UP with fire retardants, the temperature and intensity of degradation depended on FR′s type and amount. The addition of LHP led to an earlier start of the decomposition process of polymer than APP; however, the formed char is characterized by a lower intensity of decomposition under 800 °C and a higher yield of carbonaceous residue.

Water resistance was assessed by soaking the samples in water at elevated temperature and supplemented with TG analysis after their drying. Only UP and materials with the highest share of FR were analyzed, which was supposed to reflect the worst-case scenario. The obtained values are given in Appendix A, while the TG and DTG curves are presented in Figure 4.

Percentage change in mass after soaking in water at 70 °C for UP reached approx. 1.5%, while in the case of UP/APP, it was almost 4%. However, the highest value of leaching (1), several times higher than the reference materials, was obtained for polymer modified with 30 wt. % of LHP. Interestingly, only for UP/30APP, a shift toward lower temperatures in the case of T_5%_ was recorded, while for UP and UP/30LHP, the values before and after water immersion were similar. The only significant changes were the appearance of a clear peak corresponding to the degradation of pre-char for the unmodified polymer and an increase in the residue in the case of UP/LHP.

### 2.4. Mechanical Properties

DMTA is a measurement technique that allows observing basic changes in viscoelastic properties indirectly to assess both structural changes of the polymeric materials and interfacial interactions in complex polymeric systems [17]. Figure 5 shows changes in storage modulus (G’) and damping factor (tan δ) as a function of temperature for samples made of a polyester resin containing 30 wt. % of the modifier subjected or not to soak in water. The thermomechanical curves of material samples after soaking (indexed as “ai” in Figure 5 legend) are shown using a pointed line. The additional data from the DMTA, including C factor and glass transition determined as a peak of α-relaxation of UP at tan δ vs. T curve, are collectively presented in Table 2.

The introduction of both types of IFRs resulted in an increase in the stiffness of UP relative to unmodified UP over the entire considered temperature range. It should be emphasized that slightly higher G’ values were observed in the UP/30LHP than for UP/30APP sample. In turn, samples modified with IFRs after the water immersion process showed a different tendency. In the case of the UP/30APP sample, the increase in G’ relative to unmodified UP was maintained, while the values of storage modulus of the material containing 30 wt. % of LHP were decreased in some considered temperature range. This is probably due to the dissolution and washing out of the water-soluble LHP, which led to the formation of voids in the specimens revealing a decreased mechanical performance. According to the literature, polyester resins subjected to water soaking in time used in our investigations are not pronounced to significant degradation processes [18]. The storage modulus values of unmodified UP before and after glass transition are comparable. According to Fraga et al. [19], the decrease in the tan δ may be caused by styrene or monomer extraction. The shift of the peak of α-relaxation at tan δ vs. T curve, caused by water immersion, was also observed. The temperature of the water bath was the same as the applied post-curing temperature. Therefore, it cannot be excluded that the Tg increase may be related to extended post-curing processes [20] and a change of molecular weight distribution [19]. In the case of materials containing 30 wt. % IFRs, a significant increase in Tg value after water immersion was also observed. Moreover, the glass transition values for both modified series are comparable. Assuming that the addition of APP is not a typical insoluble particle dispersion reinforcement for UP, which is observed in the case of the introduction of inorganic fillers [21], it can be adopted that its presence partly affected reactivity and cross-linking behavior.

Usually, changes of storage modulus induced by the incorporation of fillers and modifiers may be correlated with elasticity modulus values determined by static tensile or flexural tests [22]. The observed increase of the G’ noted for both UP/FR series is in good agreement with literature data reporting improvement of the elasticity modulus of fire retarded thermoset polymers, including unsaturated polyester and epoxy resins [23,24]. Simultaneously, it can be supposed that the mechanical behavior of materials with 30 wt. % content of the IFR will be reduced. Considering the most popular final use of flame retarded thermoset compositions as matrices of layered composites reinforced with long fibers, changes in the properties of the matrix itself should not constitute a significant limitation in the final products’ performance [25]. Moreover, reported by SEM analysis lack of pores, the most severe structural defect [26], in the structure of compositions containing both LHP and APP allows states the high application potential of the materials.

The *υ*_e_ calculated according to the rubbery-elasticity theory in reference to Equation (2) are collectively presented in Table 2. The calculated cross-link density shows, for both modifiers, a significant increase in this structural parameter compared to unmodified UP. It can be seen that the opposite effect of water immersion in changes of *υ*_e_ may be observed. The calculated values are comparable to those presented before in the literature by Shivarkar et al. [27]. For unmodified UP the water immersion process caused a slight and negligible increase in the cross-link density, while for both IFR a serious decrease in this parameter was observed, more pronounced for UP/30 LHP. Therefore, it can be supposed that both additives influenced the cross-linking process and became rinsed out during long-time immersion in water, along with low molecular weight cross-linked products.

Analysis of the C factor thermomechanical parameter allows comparing the changes induced by the presence of different modifiers in a qualitative way. In the considered case, it can be clearly seen that the efficiency of the LHP is higher than for APP. This effect is reversed for soaked samples. The significant increase in C factor for the UP/30 LHP sample after long-term immersion is understandable if the water solubility of LHP is taken into account. An increase in C value in UP/30APP ai may be associated with partial erosion occurring on the sample surface and migration of low molecular weight products.

### 2.5. Reaction to a Small Flame

The horizontal burning (HB) test was applied to evaluate polymers’ flammability, and the results are listed in Table 3.

As can be seen, the obtained linear burning rate values, or self-extinguishing before the flame reached the first marker, allowed to include all materials in the HB class. Unmodified resin burned intensively and emitted a lot of smoke; burning drops could also be observed (Figure 6). The addition of 10 wt. % of FRs reduced the linear burning rate values as well as the intensity of accompanied phenomena, and instead of that, the formation of char was observed. In turn, the incorporation of 20 or 30 wt. % of APP and LHP led to the extinguishing of UP in a short time after the removal of the burner. The results suggest that the minimum share of FRs equals at least 20 wt. %.

### 2.6. Fire Behavior

Cone calorimeter results may be used to determine the fire properties of materials and generate data for simulating real-scale fire behavior, such as heat release rate, total heat release, or fire growth rate [28]. Table 4 presents the average values of key parameters determined for unmodified unsaturated polyester resin and UP with intumescent fire retardants, while Figure 7 presents the most representative heat release rate (HRR) curves gained during CC tests.

Time to ignition (TTI) defines the time required for the ignition of a sample and sustenance of the flame over the whole of its surface [28]. The addition of APP resulted in higher TTI values than UP, especially for materials with a higher amount of FR, while for LHP, a slightly higher value was obtained only for the UP/30LHP. The reduction of time to ignition can be associated with the lower thermal stability of the UP/LHP and is consistent with the results of the thermogravimetric analysis. Since TTI depends on the material properties such as thickness, density and thermal conductivity, it is not as reliable as other parameters, such as heat release rate [28].

HRR is considered the most critical factor for determining the fire properties [28]. The changing trend of the curve, as presented in Figure 7, shows the burning behavior of the material as a function of time. It can be observed that FRs cause a decrease in the peak HRR (pHRR) compared to unmodified resin and a flattening of the curve. PHHR values for UP reached 792 kW/m^2^, while for a polymer with 30 wt. % of APP and LHP the values were almost three (reduction by 64%) and four (reduction by 75%) times lower, respectively. Similarly, the lowest values of the maximum average rate of heat emission (MARHE), regarded as a parameter to assess the hazard of fire spread [29], were obtained for UP/30LHP.

Total heat release (THR) represents a measurement for the fire load of polymeric material [30]. In the case of investigated samples, a reduction from 159 MJ/m^2^ for the unmodified resin to 95 MJ/m^2^ for the samples containing the highest amount of FRs was noted. A decrease in THR indicates incomplete combustion due to char formation or lowered combustion efficiency [30]. Since all of the UP/FRs yielded a significant amount of residue, a THR reduction is related mainly to the char formation. This is confirmed by the residue’s growth from 9% to 29% for the UP/30LHP or even 36% for the UP/30APP. Moreover, the photographs of samples with the highest share of FRs after cone calorimeter tests (Figure 8) confirm the formation of a significant size char layer (approx. 30 mm), characteristic of intumescent fire retardants. However, a small decrease from 21 MJ/kg to 19 MJ/kg in EHC was also observed, suggesting that some activity of the organophosphorus FRs in the gas-phase occurred [30].

The morphology of the char residues after cone calorimetry tests was investigated by SEM. As can be seen in Figure 9, both formed chars have a continuous and smooth outer part; however, in the case of UP/30LHP, some bubbles suggesting porous structure inside can also be observed. A swollen structure, visible in the inner part of the residues, indicates that both the commercial as well as novel FR act as intumescent flame retardants. The chemical composition, determined by SEM-EDS, showed that the residues were comprised mainly of C, O, P and N elements. This implies that the IFRs were divided into decomposition products, including oxides (see point 3.9), leading to more compact carbonaceous char with good mechanical performance. Detected silicon atoms originated from the contaminations of the unsaturated polyester resin samples after CC tests, similar to our previous findings [31].

### 2.7. Smoke Emission

Smoke emission during a fire is a crucially important parameter due to its toxicity and reduced visibility. A few important smoke indicators, such as total smoke release (TSR) and specific extinction area (SEA), were obtained from the cone calorimetry test and are presented in Table 5. SEA corresponds to the light absorption by the particles of smoke, formed during the burning of 1 kg of the material and indirectly informs about visibility during the fire [32]. The lowest SEA was recorded for UP/30LHP, and the obtained value was 22% lower than UP. Similarly, the addition of either APP or LHP led to a gradual reduction in TSR along with the increase in FR amount. The highest reductions in TSR, by 50% and 49% for UP/30APP and UP/30LHP, respectively, were determined. The char structure, the most effective in UP with the highest share of FRs, prevented the release of incomplete combustion products, which are components of smoke.

In turn, the smoke density chamber provides information about the change in the optical density of smoke emitted from a sample exposed to the heat flux and accumulated in a closed chamber. The smoke density is calculated by measuring the light beam’s obscuration using a photosensor, while VOF 4 relates to the total optical densities measured in the first 4 min of the test [32]. The superior reduction of both parameters was obtained for UP/30LHP and Ds_max_ as well as for VOF, reaching 31% compared to the unmodified resin. Furthermore, by comparing the results obtained for samples with different amounts of APP and LHP, better results were obtained for the developed FR.

### 2.8. Characterization of Decomposition Products

The TGA/FT-IR technique was used to investigate the volatile products during the thermal decomposition process of UP, UP/30APP and UP/30LHP. Figure 10 is the three-dimensional diagram of the gas phase, and Figure 11 is the FT-IR spectra of volatilized products released at the maximum evolution rate of tested materials. As shown in Figure 11, the highest maximum weight loss for tested materials was observed after 20.06 min in the case of UP, 19.89 min for UP/30APP and 17.80 min for UP/30LHP. The time corresponds to the maximum temperature at 388, 385 and 342 °C, respectively.

In Figure 11, the representative peaks of unsaturated polyester resins are prominent in regions of around: 4000–3500 cm^−1^ (H_2_O), 3100–2700 cm^−1^ (CH, hydrocarbons), 2400–2300 cm^−1^ (CO_2_), 2200–2100 cm^−1^ (CO), 1870–1800 cm^−1^, 911–909 cm^−1^ (anhydride), 1800–1700 cm^−1^ (C=O, carbonyl compounds), 1600–1400 cm^−1^ (C=C, aromatic compounds), 1265 cm^−1^ and 1102 cm^−1^ (C-O-C, ester groups). Based on the spectra analysis of UP/LHP, no occurrence of new peaks was noted, which could indicate that substances other than those released during thermal decomposition of UP were emitted [33]. Only material UP/30APP small absorption peaks in the wavenumber 965 and 930 cm^−1^, connected with the NH_3_ from APP, are observed. The volatile gases such as NH_3_ can act as the blowing agent for expanding the intumescent char. It should be emphasized that materials with the IFRs have lower intensity of the bands.

### 2.9. Fire Retardant Mechanism

The decomposition process of LHP was described in our previous paper [34]. It was established that the decomposition process proceeds according to one of at least two identified mechanisms. The first mechanism shows that the diazole particle detaches from the amino acid and the amine group decomposes to gaseous ammonium and nitrogen oxides. Resulting linear propionic acid decomposes with the release of carbon oxides and water, providing a small heat sink effect and diluting the oxygen concentration in the sample’s immediate vicinity. Phosphoric acid dehydrates from either phosphorus pentoxide or polyphosphoric acid that can participate in the char formation process [35]. It is possible that some amount of hydrogen cyanide was formed during the decomposition; however, it was not detected during the tests [36,37].

The second mechanism assumes the inner cyclization process of LHP and the formation of a two-ring structure consisting of 2-amino-2,4-cyclopentadien-1-one and imidazole, which can co-create a char. Generating a high amount of nonflammable gases combined with a strong char structure isolates the polymer from the heat, slowing down the decomposition process and hence the burning rate of the material.

## 3. Experimental

### 3.1. Materials

The main compounds used in the synthesis of L-histidinium dihydrogen phosphate-phosphoric acid were histidine from Apollo Scientific (> 99%) (Stockport, UK) and phosphoric acid from Honeywell (85% conc.) (Charlotte, NC, USA). Unsaturated polyester resin AROPOL M 105 TB and the initiator agent BUTANOX M-50, both from Ashland Inc. (Wilmington, DE, USA), were used as a polymer. The commercial halogen-free flame retardant ammonium polyphosphate (APP) was used to manufacture a reference material.

LHP (Figure 12) was synthesized accordingly to previously reported studies [34]. Initially, 78 g (0.5 mol) of L-histidine and 100 mL H_2_O were mixed in a 1000 mL beaker, equipped with a mechanical stirrer. While stirring, the mixture was heated up to 50 °C in order to dissolve the amino acid. 115 g (approx. 1 mol) of 85% the solution of phosphoric acid was added dropwise during constant stirring, leading to an exothermic reaction, which heated the mixture to 80 °C. The reaction mixture was stirred at 80 °C for 2 more hours. The mixture was left to cool down while constantly stirred, which resulted in crystallization of the product after ca. two hours. The 400 mL of cooled methanol was added to the mixture, and it was left for another 15–20 min under constant stirring. The precipitate was filtered, washed with 50 mL of cooled methanol and left to dry. The amount of dried product was approx. 167 g.

Firstly, LHP (10 wt. %, 20 wt. % or 30 wt. %) was added to the resin and stirred using a high-speed mechanical stirrer with a water jacket. During mixing, the rotational speeds of 7000, 10,000 and 17,000 rpm were applied for 2, 1 and 0.5 min. (±10 s), respectively. The corresponding amount of initiator agent (1 wt. %) was introduced, and the mixture was stirred again for 1 min. Afterward, the UP was degassed for 7.5 min. (±10 s), poured into molds and cured at ambient temperature for approx. 24 h, as well as post-cured at 70 °C for 3 h. Next, the samples were conditioned at ambient temperature for 2 weeks. The formulations of flame retarded UP were given in Table 6. It is important to notice that the addition of LHP and APP is related to different content of phosphoric acid in a final product. In the case of 10 wt. % APP, the overall input of phosphorus is 3.2 wt. % while in the case of LHP the input of phosphorus is at the level of 1.8% (higher value of the flame retardant consistently introduces a multiple of the given phosphorous contents).

### 3.2. Methods

The Fourier transformed infrared (FT-IR) spectra were recorded using a Vertex 70 spectrometer from Bruker Corp (Billerica, MA, USA). Spectra in the range of 600–4000 cm^−1^ were obtained by 64 scans at a resolution of 4 cm^−1^.

Elemental analysis (CHN) data were collected using the Elementar Unicube analyzer (Elementar Analysensysteme, Langenselbold, Germany). Examined powders were initially dried at 120 °C for 12 h to remove any residual water. Samples of ca 3 mg were weighted and closed in tin cups directly before the analysis. Each analysis was repeated three times to evaluate the errors.

Scanning electron microscope SU8010 (Hitachi, Tokyo, Japan) was used to examine the samples’ morphology and the residues obtained from CC tests. The UP/FR was gold-coated using a Quorum Technologies Q150T ES sputter coater (Quorum, Laughton, UK) to improve the conductivity. Observations were conducted at an accelerating voltage of 5 kV and magnifications 500×. The residues did not require sputter coating and fixed on a table using LEIT-C Conductive Carbon Cement. Point elemental analysis was carried out using the Thermo Scientific NORAN System 7 equipped with an electrically cooled Silicon Drift Detector EDS detector (Thermo Scientific UltraDry, Waltham, MA, USA) to investigate the chemical composition of the residues.

The thermogravimetric analysis was carried out on a TGA 5500 series Discovery (TA Instruments Ltd., New Castle, PA, USA). The samples (10 mg) were tested in an atmosphere of nitrogen with flow gas at a rate of 10 mL/min in the chamber and 90 mL/min in the oven. Samples were heated from room temperature up to 800 °C at a rate of 10 °C/min. The initial decomposition temperature T_5%_ was defined as the temperature at which the weight loss was 5%, and the residual mass was determined at 800 °C. The maximum temperature and rate of degradation were also determined based on derivative thermogravimetric curves (DTG).

The water resistance was assessed by soaking the samples in distilled water at 70 °C for 168 h and dried at 70 °C to a constant mass using a Plus II Incubator (Gallenkamp, London, UK). The leaching of FR was calculated according to the following equation:(1)$$\mathrm{Leaching}=\frac{\left(w_{0}-w\right)}{w_{0}}\times 100\%$$
where *w*_0_ is the initial mass of the sample, and W is the sample’s mass after soaking and drying. Next, the samples were subjected to a TGA conducted in accordance with the methodology described above.

The dynamic-mechanical properties of the specimens (50 × 10 × 4 mm³) were measured using MCR 301 from Anton Paar (Graz, Austria) in torsion mode, using a frequency of 1 Hz in the temperature range of 25 °C and 200 °C and a heating rate of 2 °C/min. The position of tan δ at its maximum was taken to determine the glass transition temperature (Tg). From the rubber elasticity theory, the rubbery modulus is proportional to the cross-link density (*ν*_e_) [38,39], which can be evaluated from the following Equation (2) [40]:(2)$$\nu_{e}=\frac{E\prime_{r}}{3\phi RT}$$
where *E′_r_* is the storage modulus in the rubbery region, *R* is the gas constant (8.3145 [J·mol^−1^·K^−1^], *ϕ* is the front factor which may be assumed to unity for a good order-of-magnitude prediction, and *T* is the absolute temperature. In our studies, due to the application of DMTA measurements in torsion mode, the *G′* was used instead of *E′*. The *G′_r_* values were determined in a rubbery plateau range at a constant temperature of 140 °C. From thermomechanical, data additional parameter the effectiveness of the filler, which is based on the determination of the *C* factor defined as follows [41]:(3)$$C=\frac{{\left(G\prime_{g}/G\prime_{r}\right)}_{\mathrm{modified}\;\mathrm{polymer}}}{{\left(G\prime_{g}/G\prime_{r}\right)}_{\mathrm{polymer}}}$$
where *G′_g_* and *G′_r_* are the values of storage modulus determined in the glassy and rubbery state of the material; in the presented case, *G′* values were taken at 30 °C and 140 °C.

The horizontal burning tests were performed according to the standard IEC 60695-11-10. The specimens (125 × 10 × 4 mm³) were subjected to a 50 W test flame. The burning was assessed using its linear burning rate (v) calculated from the following equation:(4)$$v=\left(\frac{L}{t}\right)\times (\frac{60s}{\min})$$

Fire behavior was assessed using a cone calorimeter (Fire Testing Technology, East Grinstead, UK). The samples (100 × 100 × 6 mm³) were placed in aluminum foil and tested at a heat flux of 35 kW/m^2^ applied horizontally to the sample, in conformity with ISO 5660 standard. The separation space between samples and the heater was set at 25 mm. Spark ignition was used to ignite the pyrolysis products. The residues were photographed using a digital camera EOS 400 D (Canon Inc., Tokyo, Japan).

The optical density of smoke was assessed using a smoke density chamber (Fire Testing Technology Ltd., East Grinstead, UK). The samples (75 × 75 × 6 mm³) were exposed to a heat flux of 25 kW/m^2^ without applying the pilot flame, in conformity with ISO 5659-2 standard. As in the CC tests case, the values are the average obtained for three samples from each series.

The gas-phase analysis was carried out in TGA Q500 (TA Instruments Ltd., New Castle, PA, USA) coupled with FT-IR Nicolet 6700 spectrometer (Thermo Scientific, Waltham, MA, USA), using 64 counts. The samples (10 mg) were heated in the air from room temperature up to 800 °C at a rate of 20 °C/min. To reduce the possibility of evolved products condensing along the transfer line the FT-IR gas cell was held at 240 °C, and the temperature of the transfer line was set at 250 °C. The analyses were performed in a spectral range of 400–4000 cm^−1^ and with a resolution of 4 cm^−1^.

## 4. Conclusions

As part of the work, a L-histidinium dihydrogen phosphate-phosphoric acid was prepared and characterized. Based on FT-IR and CHN analyses, the composition and purity of the synthesized LHP were confirmed. Next, the substance was added to the unsaturated polyester resin to verify its effectiveness in flammability inhibition. The results were compared to those obtained for unmodified UP and resin with a commercial intumescent fire retardant.

UP modified with LHP was characterized by higher thermal stability, confirmed by a lower decomposition rate and the highest residue yield. The lowest flammability and smoke emission were noted for the UP containing 30 wt. % of LHP. It was indicated that the LHP contributed to the formation of carbonaceous char, improving the flame retardant properties of the polymer, while the studies of SEM confirmed its swollen structure. Moreover, a release of non-flammable gases was recorded, which confirmed that LHP acted both in the condensed as well as in the gas phase. The results were better compared with the UP and similar (HB) or better than (TGA, CC, smoke density chamber) a resin containing the same amount of commercial IFR.

However, an important disadvantage concerning water resistance was observed. The leaching of LHP from UP/LHP reached 15%, which affected the structure and thermomechanical properties of materials. For this reason, the new flame retardant can only be used in products that do not come into direct and long-term contact with water, especially those with increased temperature. Nonetheless, it is an interesting direction towards the development of new flame retardants.

## Figures and Tables

**Figure 1 molecules-26-00932-f001:**
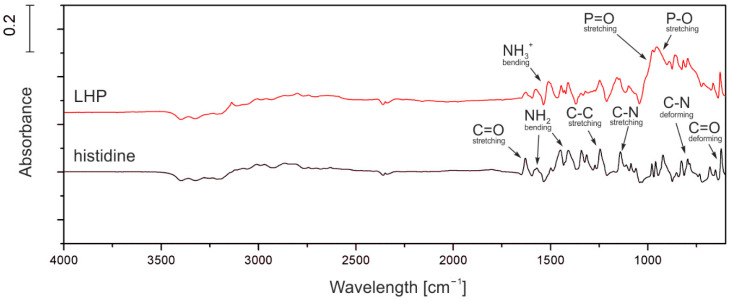
FT-IR graph of histidine and LHP.

**Figure 2 molecules-26-00932-f002:**
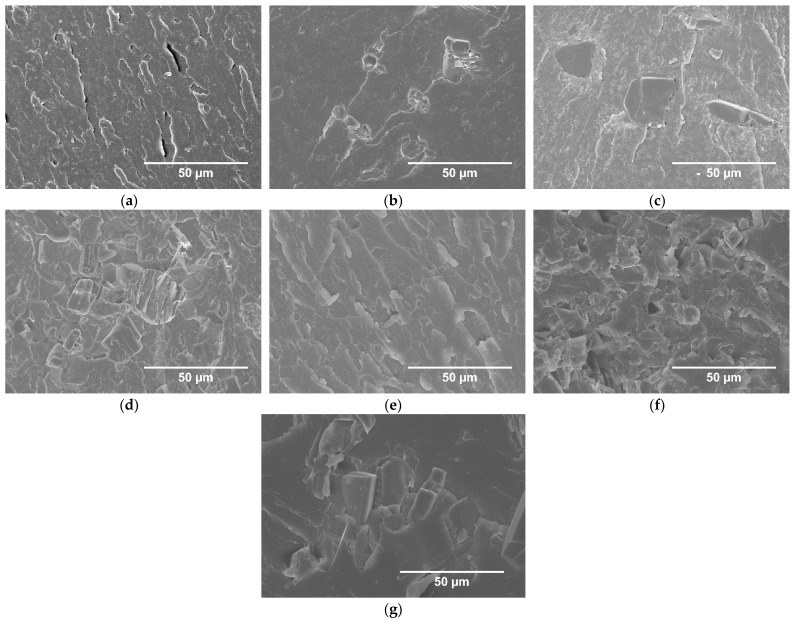
SEM images of breakthroughs of UP (**a**), UP/10 ammonium polyphosphate (APP) (**b**), UP/20APP (**c**), UP/30APP (**d**), UP/10LHP (**e**), UP/20LHP (**f**), UP/30LHP (**g**).

**Figure 3 molecules-26-00932-f003:**
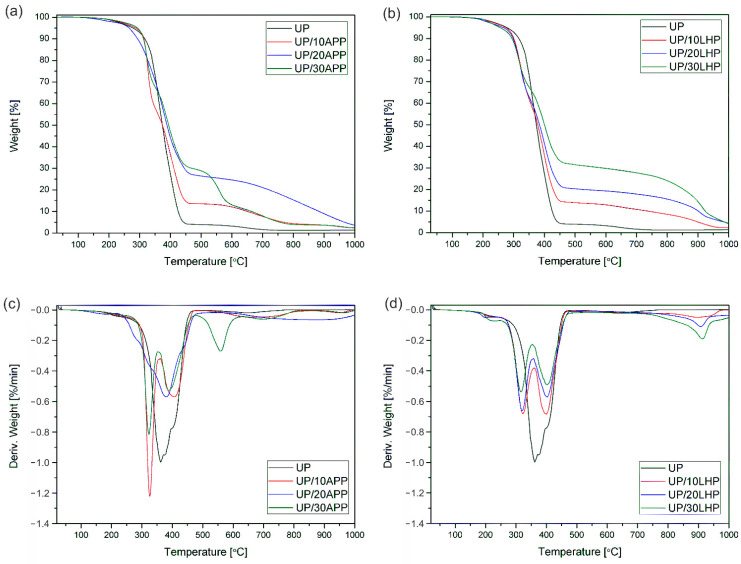
TG and DTG curves of UP and UP/APP (**a**,**c**) as well as UP/LHP (**b**,**d**) under an inert atmosphere.

**Figure 4 molecules-26-00932-f004:**
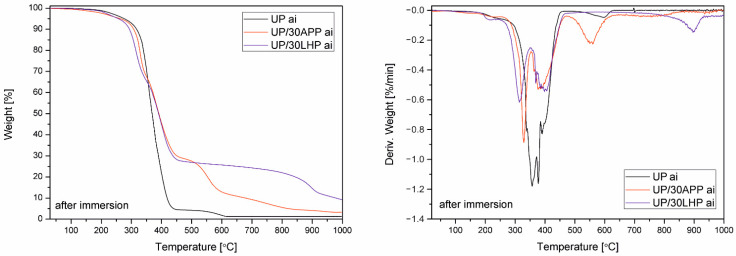
TG and DTG curves of UP and APP and LHP under an inert atmosphere after water soaking.

**Figure 5 molecules-26-00932-f005:**
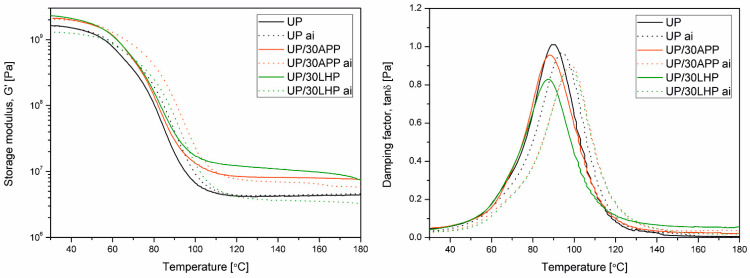
DMA results of unmodified and modified with APP and LHP before and after immersion in water (ai).

**Figure 6 molecules-26-00932-f006:**
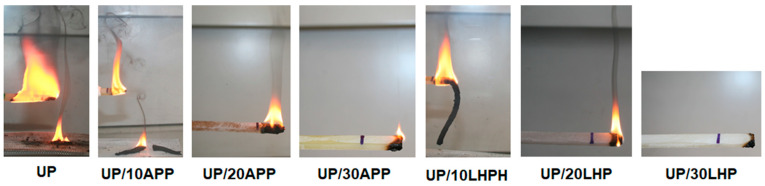
Photographs of the UP and UP/IFRs during the horizontal burning tests.

**Figure 7 molecules-26-00932-f007:**
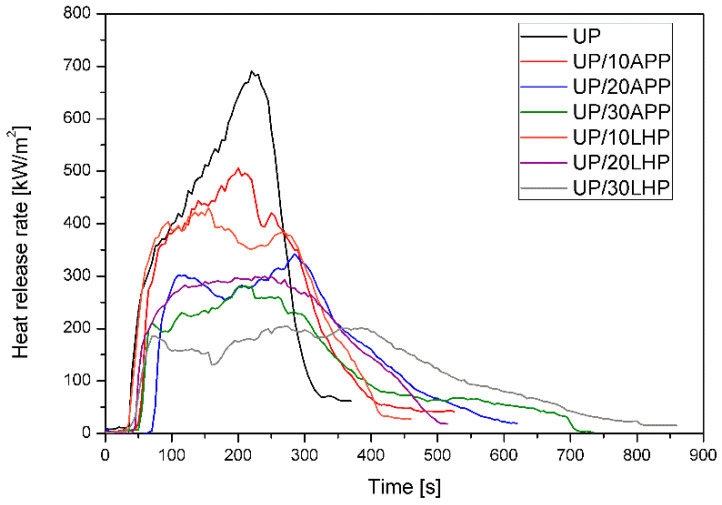
Representative curves of the heat release rate of the UP and UP/IFRs.

**Figure 8 molecules-26-00932-f008:**
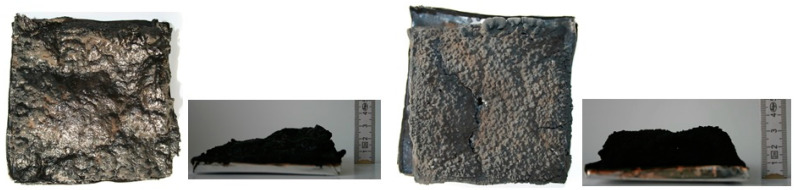
Photographs of the UP/30APP and UP/30LHP after cone calorimetry tests.

**Figure 9 molecules-26-00932-f009:**
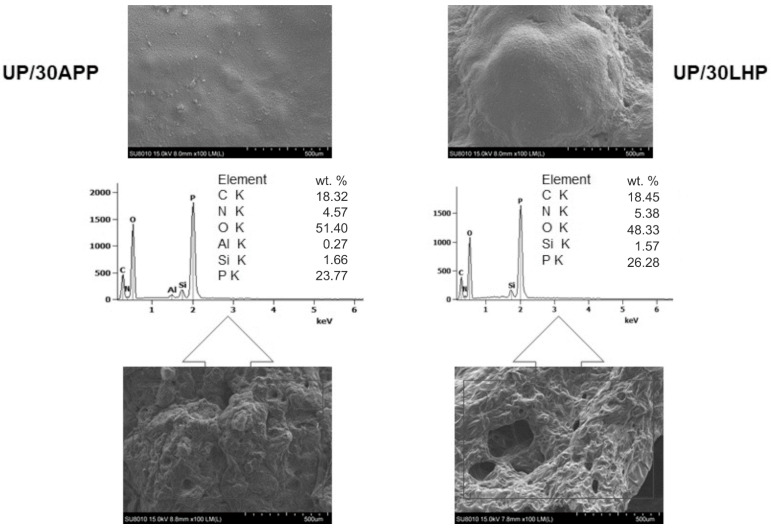
SEM images of UP/30APP and UP/30LHP after CC tests and EDS results (outer and inner part of char).

**Figure 10 molecules-26-00932-f010:**
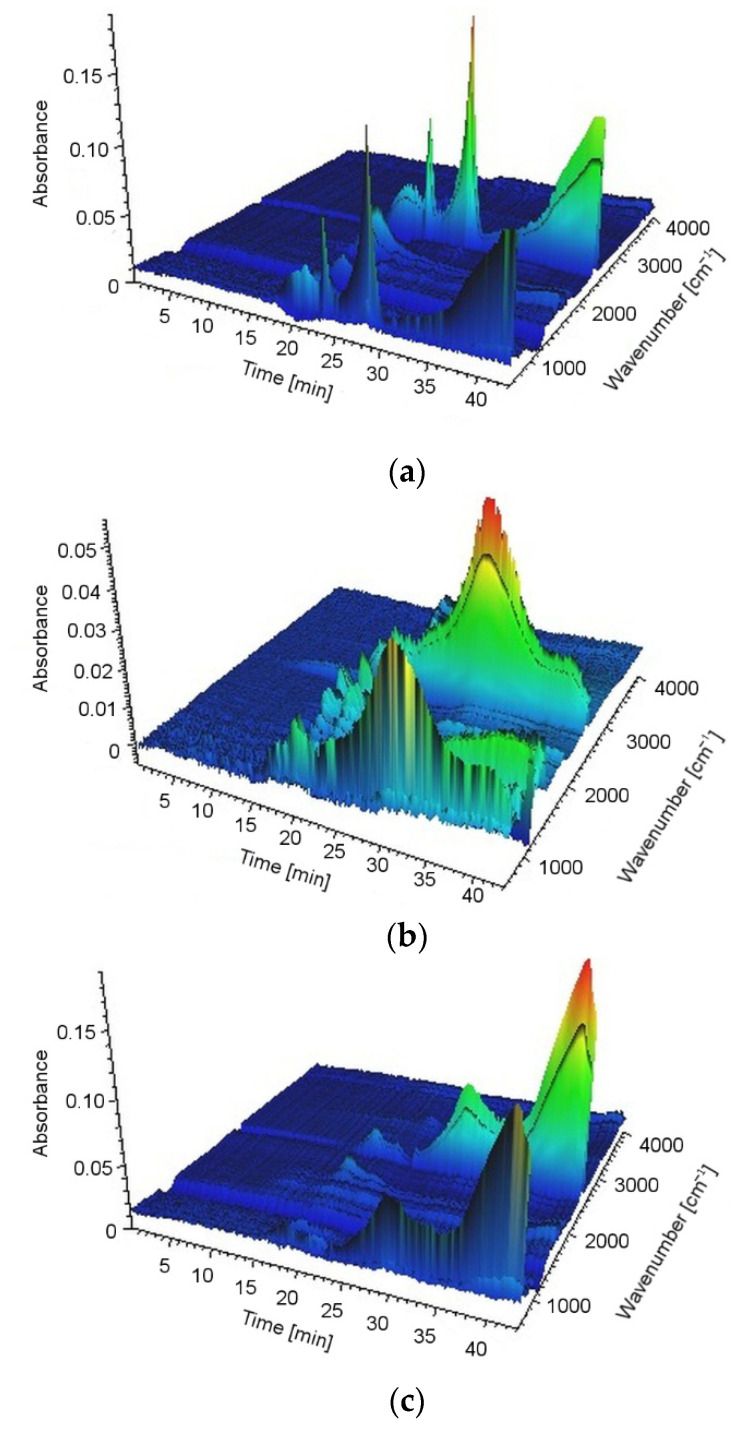
The 3D-IR spectra of UP (**a**), UP/30APP (**b**) and UP/30LHP (**c**).

**Figure 11 molecules-26-00932-f011:**
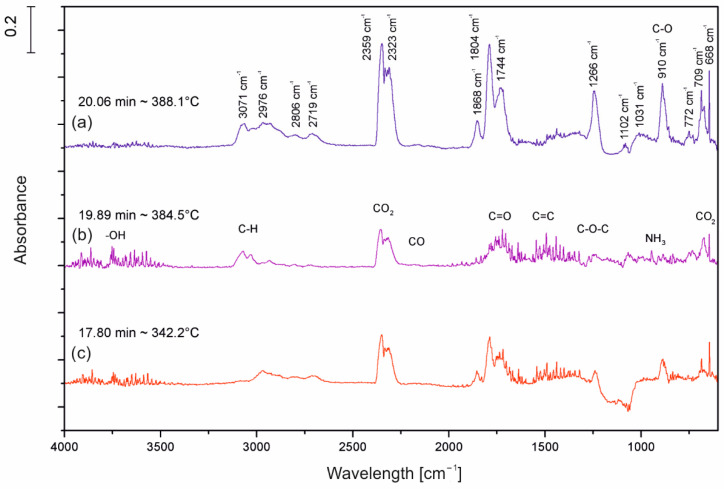
FT-IR spectra of volatilized products released at the maximum evolution rate from UP (**a**), UP/30APP (**b**) and UP/30LHP (**c**).

**Figure 12 molecules-26-00932-f012:**
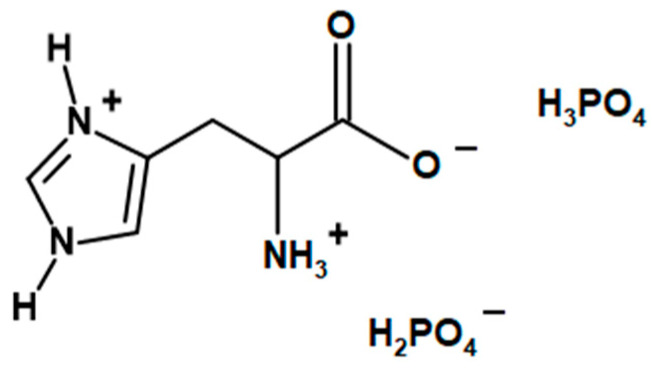
The chemical structure of L-histidinium dihydrogen phosphate-phosphoric acid (LHP).

**Table 1 molecules-26-00932-t001:** Elemental analysis of LHP.

Samples	Mass, %		
	C	H	N
LHP (experiment)	21.53 ± 0.05	4.28 ± 0.08	12.45 ± 0.05
100% LHP (calculated)	20.52	4.31	11.97
96% LHP + 4% histidine (calculated)	21.56	4.37	12.57

**Table 2 molecules-26-00932-t002:** Structural and thermomechanical properties of unmodified and modified UP before and after water immersion obtained by DMTA.

Materials	UP	UP ai	UP/30APP	UP/30APP ai	UP/30LHP	UP/30LHP ai
Tg, °C	90.2	94.0	88.3	98.4	87.4	97.7
*υ*_e_, ×10^−3^ (mol·cm^−3^)	0.411	0.421	0.789	0.675	1.058	0.355
C	-	-	0.674	0.790	0.48	0.957

ai—after water immersion

**Table 3 molecules-26-00932-t003:** Horizontal burning test of unmodified UP and resin with APP and LHP.

Materials		UP	UP/	UP/	UP/	UP/	UP/	UP/
			10APP	20APP	30APP	10LHP	20LHP	30LHP
The linear burning rate, mm/min	I	22.4	6.8	-	-	12.0	-	-
	II	27.3	10.4	-	-	9.7	-	-
	III	23.6	9.8	-	-	12.0	-	-
Classification	-	HB	HB	HB	HB	HB	HB	HB

**Table 4 molecules-26-00932-t004:** Cone calorimeter results of UP and UP/IFRs.

Materials	UP	UP/	UP/	UP/	UP/	UP/	UP/
		10APP	20APP	30APP	10LHP	20LHP	30LHP
TTI, s	50 (10) ^a^	51 (4)	72 (9)	58 (4)	41 (3)	46 (4)	54 (7)
pHRR, kW/m^2^	792 (27)	544 (36)	332 (20)	283 (20)	413 (15)	314 (32)	200 (5)
MARHE, kW/m^2^	459 (51)	353 (22)	218 (14)	196 (7)	310 (9)	233 (9)	152 (7)
THR, MJ/m^2^	159 (34)	116 (4)	102 (3)	95 (5)	120 (6)	102 (4)	95 (3)
EHC, MJ/kg	21 (1)	19 (1)	19 (1)	21 (1)	20 (1)	20 (1)	19 (1)
Residue, %	9 (1)	13 (0)	21 (1)	36 (2)	14 (0)	24 (0)	29 (0)

^a^ The values in parentheses are the standard deviations.

**Table 5 molecules-26-00932-t005:** Smoke emission of UP and UP/IFRs.

Materials	UP	UP/	UP/	UP/	UP/	UP/	UP/
		10APP	20APP	30APP	10LHP	20LHP	30LHP
SEA, m^2^/kg	918 (44) ^a^	967 (12)	903 (55)	770 (59)	935 (6)	823 (30)	716 (54)
TSR, m^2^/m^2^	6895 (1582)	5882 (64)	4833 (288)	3445 (210)	5625 (55)	4804 (138)	3535 (291)
Ds_max_	1068 (35)	973 (41)	923 (66)	790 (31)	934 (11)	818 (154)	733 (8)
VOF4	313 (77)	396 (92)	404 (84)	309 (27)	332 (4)	204 (21)	216 (34)

^a^ The values in parentheses are the standard deviations.

**Table 6 molecules-26-00932-t006:** Formulations of flame retardancy unsaturated polyester resin (UP).

Samples	Components, wt. %		
	UP	APP	LHP
UP	100	0	0
UP/10APP	90	10	0
UP/20APP	80	20	0
UP/30APP	70	30	0
UP/10LHP	90	0	10
UP/20LHP	80	0	20
UP/30LHP	70	0	30

## Data Availability

The data presented in this study are available on request from the corresponding author.

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
