# Peer review of "Moisture Resistance, Thermal Stability and Fire Behavior of Unsaturated Polyester Resin Modified with L-histidinium Dihydrogen Phosphate-Phosphoric Acid"

_molecules, 2021, doi:10.3390/molecules26040932_

Round 1

Reviewer 1 Report

This fundamental research work mainly introduces the impact of a new flame retardant on the thermal stability and flammability of unsaturated polyester resin. It is an interesting work but also needs to be approved in some aspects. Here are some advises that are hoped to improve this work and reach the publication requirement of Molecules.

  • The abstract and conclusion parts are really too many words to grab the key idea and findings of this work, the authors should let the long story short.
  • As stated by authors that after loading of 30%, LHP could significantly reduce the HRR value near 75%. I suppose if this kind of high loading could worse the mechanical properties and long-term ageing behaviour.
  • Here is an advice, if there is a figure, such as TGA, Cone test results, that have already described the results, a repeated table data should be placed in Supporting Info, which could make the manuscript more concise and concrete. 
  • All the figures should be re-designed and reformatted due to some issues, such as font, unclear words, etc. These curves are unprofessional. 

Author Response

We would like to thank the Reviewer for all the valuable remarks. We agree with the recommendations that the manuscript should be improved and so the effort has been made to correct the article according to the comments.

  1. The abstract and conclusion parts are really too many words to grab the key idea and findings of this work, the authors should let the long story short.

Thanking the Reviewer for the remarks we would like to say that the suggested changes have been made.

  1. As stated by authors that after loading of 30%, LHP could significantly reduce the HRR value near 75%. I suppose if this kind of high loading could worse the mechanical properties and long-term ageing behaviour.

One of the aims of this study was to describe the behavior of the UP modified with LHP as well as potential limitations resulting from solubility in the water of the modifier. The presented research results constituted preliminary works and were to determine the potential hazards resulting from the physicochemical properties of LHP, which will determine its future applications. The thermomechanical properties using the DMA method were included in the research to confirm the proper realization of the curing process as well as pre-check the course of mechanical properties changes. Unfortunately, the number of materials produced in the first series did not allow for the realization of complex mechanical properties analyses.

The Reviewer rightly noticed that the mechanical properties might change as a result of the introduction of a monomer-insoluble powder modifier, which constitutes a hindrance in the polymer structure, comparable to the particle-shaped filler. According to the literature addition of IFR in a solid-state in the form of powder into thermoset polymers such as unsaturated polyester or epoxy resin led to the deterioration of its mechanical properties. As reported by Gao et al. results showed about 50% decrease of tensile and flexural strength, with a simultaneous 25% increase of elasticity modulus of the UP with 30 wt% of the APP in comparison to neat UP [10.1002/app.49148]. Cheng and Kuo studied the influence of APP and hyperbranched silicon-containing polymers on fire behavior and mechanical properties of epoxy resin [10.1002/APP.48857]. Presented by them, results also showed that the addition of various IFR might reduce the strength at the break to about 50% of the unmodified polymer. However, it should be noticed that investigated UP/LHP probably be used to develop structural composites reinforced with long inorganic fibers. While the addition of IFR in the form of powder reduce the mechanical performance of thermoset polymer even by 50%, in the case of using the same composition of the modified polymer as a matrix for manufacturing of the laminates, the deterioration of the mechanical properties became almost negligible (less than 5%) as it was reported by Rajei et al. [10.1016/j.compositesb.2017.01.039]. The research works assumed in the further part of the project involve the synthesis of larger amounts of the modifier and composites' production in the form of laminates along with a complete analysis, including long-term aging.

At the same time, it should be added that in most cases, changes of the thermomechanical parameters, namely the storage modulus of the modified materials, are in line with the changes in the elasticity modulus determined using static tensile or flexural tests. Thus, the observed increase in the storage modulus of non-immersed materials containing 30 wt% of the IFR is consistent with the literature data reporting each time an increase in Young's modulus of UP modified with FR. Additional comment according to the presented explanation has been included in a revised version of the manuscript.

  1. Here is an advice, if there is a figure, such as TGA, Cone test results, that have already described the results, a repeated table data should be placed in Supporting Info, which could make the manuscript more concise and concrete.

We thank the Reviewer for precious advice and agree that the manuscript should be more concise. Bearing in mind the Reviewer's remark, we moved the TGA results to the Supporting Information section. However, in the authors' opinion, the cone calorimeter data should be placed in the main part. The average values of THR, EHC, and residue are impossible to read from HRR curves and given together with the figure better show how the applied flame retardant affects materials' fire behaviour.

  1. All the figures should be re-designed and reformatted due to some issues, such as font, unclear words, etc. These curves are unprofessional.

We thank the Reviewer for pointing that out. The Figures presenting graphs have been checked, modified, and redraw in the same manner, including corrections of legends and fonts. 

We have also corrected quite a number of other minor errors which we noticed while working on the text, and we believe that our manuscript in the present form can be published in the Journal.

Yours faithfully,

Kamila Salasinska,

Maciej Celiński,

Mateusz Barczewski

Michał K. Leszczyński,

Paweł Kozikowski

Reviewer 2 Report

Authors reported an interesting study of a new unsatured resin mixed with a phosphate flame retardant.

This paper is well written and very significant for the field.

Nonetheless, it somethimes lacks in presenting teh picture in a attractive way.

Few poitn that should be fixed:

line 46: Authors should added ref. "Bartoli, M., Rosi, L. and Frediani, M., 2019. Synthesis and Applications of Unsaturated Polyester Composites. In Unsaturated Polyester Resins (pp. 579-598). Elsevier." that provided an overview on polymerization of unsatured resins.

lines 215-224: IR spectra are poorly commented. A more detailed description of the peaks should be included in teh manuscript. Figure 2 is of very poor quality and it must be improved by marking each paek with teh correct IR mode assignation as they did for figure 12.

line 242, Figure 3: Scales are totally unreadable. Authors must improved teh picture quality.

Additionally, i suggest to adda small section for comaore the results obtained with current literature.

In the end, this is an absolutely valid work and it could be published after minor revisions.

Author Response

We highly appreciate all the comments and find them very useful. We agree with the recommendations that the manuscript should be improved, so efforts have been made to correct the article according to the comments.

  1. line 46: Authors should added ref. "Bartoli, M., Rosi, L. and Frediani, M., 2019. Synthesis and Applications of Unsaturated Polyester Composites. In Unsaturated Polyester Resins (pp. 579-598). Elsevier." that provided an overview on polymerization of unsatured resins. 

We thank the Reviewer for the valuable comment. We agree with the recommendation and, based on the provided article, carefully improved that inaccuracy in the revised version of the manuscript.

  1. lines 215-224: IR spectra are poorly commented. A more detailed description of the peaks should be included in teh manuscript. Figure 2 is of very poor quality and it must be improved by marking each paek with teh correct IR mode assignation as they did for figure 12.

The authors want to thank the Reviewer for drawing attention to the gaps in the description of FT-IR results. The suitable corrections in the figure according to the comment have been introduced. Moreover, the interpretation was significantly extended.

  1. line 242, Figure 3: Scales are totally unreadable. Authors must improved teh picture quality. 

Scale bars on the SEM images of breakthroughs have been corrected to make them more visible.

  1. Additionally, i suggest to adda small section for comaore the results obtained with current literature.

We want to thank the Reviewer for this valuable comment. However, bearing in mind the remarks from other Reviewers, to make the manuscript more concise and concrete, we decided not to increase the article's volume. According to the suggestion, the authors increased the number of references to the literature in the introduction and the research part. This article focuses on optimizing the synthesis of LHP and its amount in unsaturated polyester resin. Nonetheless, a comprehensive comparison with the commercial intumescent flame retardant (APP) was deliberately applied. Thanking the Reviewer for the remarks, we would like to say that the comment inspired us to prepare a separate article, comparing the results obtained for the developed FRs with those available in the literature.

We have also corrected quite a number of other minor errors which we noticed while working on the text, and we believe that our manuscript in the present form can be published in the Journal.

Yours faithfully,

Kamila Salasinska,

Maciej Celiński,

Mateusz Barczewski

Michał K. Leszczyński,

Paweł Kozikowski

Reviewer 3 Report

In general a quite interesting manuscript, with essential failures. The manuscript needs a careful revision. In some parts it is prepared quite lazy e.g. missing subscript in chemical formulars and so on ….

Abstract: Get a message out, there are too much details in the abstract. The measurement methods are not relevant, what is the core message of your results?

Solution (V) ???

0.5 mole? How do you use a mole in synthesis?

Tab. 1 UP/10APP twice in the description

I miss a mechanical characterization, how is the tensile strength of the material, how is the flame retardant polymer interaction?

Fig. 2. IR spectra are plotted from 4000 to 600 cm-1 and not the other way, what is the red line and the black?

By the way, if you described the synthesis before, why do you need to put the characterization of the FR in the main part, this would belong to the supporting information.

Sorry guys, if you want to tell me that the composition and purity is proven by your elemental analysis, that is quite horrible. A normal limit is a deviation of 0.4 %.

TGA: UP 30 % APP why does this curve has a decomposition at about 600 °C? And how is it possible, that with 30 % APP you just obtain a residue of 4 % ? This measurement is quite strange! Also the 20 % APP is strange, the decomposition at 320 °C is missing.

Measure ICP to determine the effective FR lose in the material!  

I finished reading the manuscript in the thermal analysis. The manuscript is not good to read, and with inclusions like this:

“The L obtained for the UP reached approx. 1.13%,“

Which L ?!

„a temperature above 550°C (DTG4-6)“

DTG? Yes I found it later in the Tab.

Author Response

We would like to thank the Reviewer for all his valuable remark. We agree with the recommendations that the manuscript should be improved, so efforts have been made to correct the article according to the comments.

  1. The manuscript needs a careful revision. In some parts it is prepared quite lazy e.g. missing subscript in chemical formulars and so on.  

Thank the Reviewer for valuable comment. We carefully improved that inaccuracy in the manuscript.

  1. Abstract: Get a message out, there are too much details in the abstract. The measurement methods are not relevant, what is the core message of your results?

We appreciate the Reviewer's remarks. The abstract has been changed accordingly.

  1. Solution (V) ???

The authors wish to thank the Reviewer for drawing attention to the errors that were made. Identified deficiencies have been corrected in the text.

  1. 5 mole? How do you use a mole in synthesis?

We thank the Reviewer for comment. We carefully improved that inaccuracy in the manuscript. It is typical to use "mol" in the synthesis, which allows running the reaction with stoichiometric proportions regardless of the substrates' mass.

  1. 1 UP/10APP twice in the description.

The authors wish to thank the Reviewer for drawing attention to the error that was made.

  1. I miss a mechanical characterization, how is the tensile strength of the material, how is the flame retardant polymer interaction?  

One of the aims of this study was to describe the behavior of the UP modified with LHP as well as potential limitations resulting from solubility in the water of the modifier. The presented research results constituted preliminary works and were to determine the potential hazards resulting from the physicochemical properties of LHP, which will determine its future applications. The thermomechanical properties using the DMA method were included in the research to confirm the proper realization of the curing process as well as pre-check the course of mechanical properties changes. Unfortunately, the number of materials produced in the first series did not allow for the realization of complex mechanical properties analyses.

The Reviewer rightly noticed that the mechanical properties might change as a result of the introduction of a monomer-insoluble powder modifier, which constitutes a hindrance in the polymer structure, comparable to the particle-shaped filler. According to the literature addition of IFR in a solid-state in the form of powder into thermoset polymers such as unsaturated polyester or epoxy resin led to the deterioration of its mechanical properties. As Gao et al., results showed about 50% decrease of tensile and flexural strength, with a simultaneous 25% increase of elasticity modulus of the UP with 30 wt% of the APP compared to neat UP [10.1002/app.49148]. Cheng and Kuo studied the influence of APP and hyperbranched silicon-containing polymers on fire behavior and mechanical properties of epoxy resin [10.1002/APP.48857]. Presented by them results also showed that the addition of various IFR might reduce the strength at the break to about 50% of the unmodified polymer. However, it should be noticed that investigated UP/LHP probably will be used for the development of structural composites reinforced with long inorganic fibers. While the addition of IFR in the form of powder reduce the mechanical performance of thermoset polymer even by 50%, in the case of using the same composition of the modified polymer as a matrix for manufacturing of the laminates, the deterioration of the mechanical properties became almost negligible (less than 5%) as it was reported by Rajei et al. [10.1016/j.compositesb.2017.01.039]. The research works assumed in the further part of the project involve the synthesis of larger amounts of the modifier and composites' production in the form of laminates along with a complete analysis, including long-term aging.

At the same time, it should be added that in most cases, changes of the thermomechanical parameters, namely the storage modulus of the modified materials, are in line with the changes in the elasticity modulus determined using static tensile or flexural tests. Thus, the observed increase in the storage modulus of non-immersed materials containing 30 wt% of the IFR is consistent with the literature data reporting each time an increase in Young's modulus of UP modified with FR. Additional comment according to the presented explanation has been included in a revised version of the manuscript.

  1. 2. IR spectra are plotted from 4000 to 600 cm-1 and not the other way, what is the red line and the black?

We thank the Reviewer for the suggestion. The suitable corrections in the figure according to the comment have been introduced.

  1. By the way, if you described the synthesis before, why do you need to put the characterization of the FR in the main part, this would belong to the supporting information. 

We thank the Reviewer for precious advice and agree that the manuscript should be more concise. Bearing in mind the remarks from all Reviewers, we moved some results to the Supporting Information section, and descriptions concerning FT-IR as well as CHN analysis have been improved. However, in the authors' opinion, the LHP characterization should be placed in the main part.

  1. Sorry guys, if you want to tell me that the composition and purity is proven by your elemental analysis, that is quite horrible. A normal limit is a deviation of 0.4 %.

We agree that the reported elemental analysis results involved too high error margins, which resulted from technical difficulties during the measurements. Therefore, we have carefully repeated the analysis, which gave results with much smaller error margins (0.08% and lower, see the revised manuscript for details). As it turned out, the improved elemental analysis values indicate that the prepared material is not 100% phase pure LHP, but involves 4% of unreacted histidine, which was explained in the manuscript.

  1. TGA: UP 30 % APP why does this curve has a decomposition at about 600 °C? And how is it possible, that with 30 % APP you just obtain a residue of 4 % ? This measurement is quite strange! Also the 20 % APP is strange, the decomposition at 320 °C is missing.

We thank the Reviewer for the comment and agree that the TG analysis results for UP/APP are unusual. The influence of the intensive swelling of the samples during the measurements, which hindered the proper course of the analyzes, cannot be excluded, especially in the case of samples with the highest IFR content. According to the suggestion, all of the parameters have been checked and corrected. Moreover, to better explain the decomposition process, the TG analysis was elongated to 1000°C, and the suitable changes have been introduced into a revised version of the manuscript. The authors want to thank the Reviewer for drawing attention to the lack of information about the decomposition of UP/20APP at approx. 300°C.

  1. Measure ICP to determine the effective FR lose in the material!

We have to admit that the Reviewer is right, and in fact, it would be advisable to quantify the content of migrated components using appropriately mass spectrometry methods. Unfortunately, we currently do not have the research material in the form of water after soaking, allowing us to perform appropriate analyses. Besides, we do not have the appropriate apparatus in our laboratories, and due to the limitations caused by the current epidemic situation, as well as the financial schedule of the project, we are unable to outsource IPC analysis. Future works that concern full accelerated weathering tests supplemented by the description of migration kinetics will be performed. We are not neglecting the proposed methodology, and definitely, it seems for us to be interesting.

  1. I finished reading the manuscript in the thermal analysis. The manuscript is not good to read, and with inclusions like this:

We are thanking the Reviewer for calling our attention to the issue. The authors would like to say that every effort has been made to improve the results' description.

We have also corrected quite a number of other minor errors which we noticed while working on the text, and we believe that our manuscript in the present form can be published in the Journal.

Yours faithfully,

Kamila Salasinska,

Maciej Celiński,

Mateusz Barczewski

Michał K. Leszczyński,

Paweł Kozikowski

Round 2

Reviewer 1 Report

Authors have revised the manuscript well and answered the questions, I suggest this work could be accepted.

Author Response

The authors wish to thank the Reviewer for all his valuable remark. 

Yours faithfully,

Kamila Salasinska,

Maciej Celiński,

Mateusz Barczewski

Michał K. Leszczyński,

Paweł Kozikowski

Reviewer 3 Report

phosphoric acid (V) why the five? The name shows that it has oxidation state of 5!!!

The manuscript improved, but the results show, that the material is not a better flame retardant as APP.

A horizontal burning test, is not significant for any application. E&E at least a vertical flame test with 0.8 or 1.6 mm material thickness, and an UL 94 rating of V1 or V0!!!

The comparison of APP to the used material just on weight % is a horrible choice, a proper choice would be the same molar ratio of P for the comparison!

Author Response

We highly appreciate all the comments and find them very useful. We agree with the recommendations that the manuscript should be improved, so efforts have been made to correct the article according to the comments.

  1. phosphoric acid (V) why the five? The name shows that it has oxidation state of 5!!! 

We thank the Reviewer for the precious advice. According to the literature, there is nothing wrong with the addition of the oxidation state after the name of the acid. This procedure eliminates the error of reading the acid name, which is crucial, especially in the case of synthesis description. It is also a common practice used in our laboratory. However, we will delete this entry to maintain the consistency of this publication.

  1. The manuscript improved, but the results show, that the material is not a better flame retardant as APP.

The aim of the work was to develop a new additive with better or at least similar properties to the best markets solutions. APP is one of the most effective and popular intumescent fire retardants. Apart from the samples with APP, the samples with melamine cyanurate (MC) were also prepared and explored (please see table below). The poorer results obtained for UP/MC prompted the authors to continue research for UP/APP as reference materials. The authors deliberately presented a critical attitude towards the solution they developed. Although LHP reduced fire spread and smoke emission, its loss during prolonged soaking in water at elevated temperature may be a weakness. We are currently working on modifying the LHP and combining it with other additives to develop flame-retardant systems, which brings interesting results. However, we believe that the obtained data may be interesting and useful for other research groups.

Materials

pHRR,

kW/m2

THR,

MJ/m2

MARHE,

kW/m2

EHC

MJ/kg

UP

792 (27)

159 (34)

459 (51)

21 (1)

UP/10MC

690 (15)

148 (1)

437 (16)

22 (0)

UP/20MC

660 (35)

144 (2)

398 (7)

22 (0)

UP/30MC

592 (68)

135 (5)

348 (30)

21 (0)

UP/10LHP

413 (15)

120 (6)

310 (9)

20 (1)

UP/20LHP

314 (32)

102 (4)

233 (9)

20 (1)

UP/30LHP

200 (5)

95 (3)

152 (7)

19 (0)

  1. A horizontal burning test, is not significant for any application. E&E at least a vertical flame test with 0.8 or 1.6 mm material thickness, and an UL 94 rating of V1 or V0!!! 

According to the standard IEC 60695-11-10, the investigation covers the samples in the range of 0.02-13.0 mm. We agree that a vertical flame test with a 0.8 or 1.6 mm samples thickness would be more useful for assessing the materials industrial applications.

The presented results are primarily of a cognitive nature and were intended to show the similarities or differences in the burning behaviour of UP with the developed LHP and commercial intumescent fire retardant. The article intentionally includes photos taken during UL-94 tests, and the use of significant thickness samples and their horizontal arrangement was supposed to show the differences in the behaviour of materials containing a similar share of FRs.  

UP/LHP will be used to develop structural composites reinforced with long inorganic fibres. The research works assumed in the further part of the project involve the synthesis of larger amounts of the developed modifiers and composites' production in the form of laminates and complete analysis, including further burning behaviour analysis. We thank the Reviver for your valuable advice, which we will use in future research.

  1. The comparison of APP to the used material just on weight % is a horrible choice, a proper choice would be the same molar ratio of P for the comparison!

We thank the Reviewer for this idea. In the materials section, we have added the information regarding the overall input of phosphorus into a final resin based on the used flame retardant and its content.

We have also corrected quite a number of other minor errors which we noticed while working on the text, and we believe that our manuscript in the present form can be published in the Journal.

Yours faithfully,

Kamila Salasinska,

Maciej Celiński